# 4D-imaging of drip-line radioactivity by detecting proton emission from $^{54m}$Ni pictured with ACTAR TPC

J. Giovinazzo [1✉], T. Roger[2], B. Blank[1], D. Rudolph [3], B. A. Brown[4], H. Alvarez-Pol [5], A. Arokia Raj[6], P. Ascher[1], M. Caamaño-Fresco [5], L. Caceres[2], D. M. Cox[3], B. Fernández-Domínguez[5], J. Lois-Fuentes[5], M. Gerbaux[1], S. Grévy [1], G. F. Grinyer [7], O. Kamalou[2], B. Mauss[8], A. Mentana[6], J. Pancin[2], J. Pibernat[1], J. Piot[2], O. Sorlin[2], C. Stodel [2], J.-C. Thomas[2] & M. Versteegen [1]

Proton radioactivity was discovered exactly 50 years ago. First, this nuclear decay mode sets the limit of existence on the nuclear landscape on the neutron-deficient side. Second, it comprises fundamental aspects of both quantum tunnelling as well as the coupling of (quasi) bound quantum states with the continuum in mesoscopic systems such as the atomic nucleus. Theoretical approaches can start either from bound-state nuclear shell-model theory or from resonance scattering. Thus, proton-radioactivity guides merging these types of theoretical approaches, which is of broader relevance for any few-body quantum system. Here, we report experimental measurements of proton-emission branches from an isomeric state in $^{54m}$Ni, which were visualized in four dimensions in a newly developed detector. We show that these decays, which carry an unusually high angular momentum, $\ell = 5$ and $\ell = 7$, respectively, can be approximated theoretically with a potential model for the proton barrier penetration and a shell-model calculation for the overlap of the initial and final wave functions.

[1] Centre d'Etudes Nucléaires de Bordeaux Gradignan, UMR 5797 CNRS/IN2P3 - Université de Bordeaux, Gradignan, Cedex, France. [2] Grand Accélérateur National d'Ions Lourds, CEA/DRF-CNRS/IN2P3, B.P. 55027, Caen, Cedex, France. [3] Department of Physics, Lund University, Lund, Sweden. [4] Department of Physics and Astronomy and National Superconducting Cyclotron Laboratory, Michigan State University, East Lansing, MI, USA. [5] IGFAE and Dpt. de Física de Partículas, Univ. of Santiago de Compostela, Santiago de Compostela, Spain. [6] Instituut voor Kern- en Stralingsfysica, KU Leuven, Leuven, Belgium. [7] Department of Physics, University of Regina, Regina, SK, Canada. [8] RIKEN Nishina Center, Wako, Saitama, Japan. ✉email: giovinaz@cenbg.in2p3.fr

Nuclear stability is governed by the underlying nuclear shell structure. Closed nuclear shells, so-called magic numbers of protons (Z) and neutrons (N), are a central concept of nuclear structure[1,2]. They confer to the nuclei a particular stability with respect to their neighbours. The study of nuclei in the vicinity of those with a magic proton number and a magic neutron number, i.e., doubly magic nuclei, is of prime interest: they allow us to adjust parameters of the nuclear shell model, a prime model at hand for the description of the structure of the atomic nucleus.

For the case of doubly magic $N = Z$ nuclei, another fundamental concept of nuclear structure physics comes into play: isospin. Isospin was introduced by Heisenberg[3] to treat the proton and the neutron as two different quantum states of a single particle, the nucleon, to acknowledge the fact that protons and neutrons have similar properties: The nucleon–nucleon interaction is to a large extent charge independent and charge symmetric. However, a closer look revealed that the Coulomb interaction and parts of the strong force violate this isospin symmetry (e.g[4]).

Nuclear physics research continues to thrive on identifying nuclei at the limits of nuclear existence in terms of Z, N, and mass number, A. Here, nuclear structure phenomena can often be filtered out more purely. Like α decay, proton emission is typically described as a quantum tunnelling process from a quasi-bound quantum state, confined for a finite time by the Coulomb and centrifugal barrier. Thus, nuclei or nuclear states beyond the proton dripline are ideal candidates to study the influence of the nuclear continuum. Furthermore, the time-reversed process of radiative proton capture is highly relevant for the synthesis of elements in certain stellar scenarios (see e.g.[5–7]), with the doubly magic nucleus $^{56}$Ni being a seed nucleus for the rapid proton capture process.

Proton radioactivity (for the latest reviews see[8,9]) was discovered 50 years ago[10], interestingly from an excited isomeric state, namely $^{53m}$Co, located at 3.19 MeV excitation energy with a spin and parity of 19/2⁻. In this decay, the proton has to carry away impressive 9ħ units of angular momentum, $\ell = 9$, to decay into $^{52}$Fe, with only a 1.5% branch to its 0⁺ ground state known. In the same region of the nuclear chart, two-proton radioactivity

was discovered from the ground states of $^{45}$Fe[11,12], $^{48}$Ni[13,14] and $^{54}$Zn[15,16].

As just mentioned, these dripline phenomena can also arise from excited states of nuclei closer to stability[17,18], in particular, if these states are relatively long-lived, i.e., isomeric like in $^{53m}$Co[10]. A second case of proton radioactivity from an isomer near $^{56}$Ni was discovered in $^{54m}$Ni[19], the mirror partner of the well-studied 10⁺ isomer $^{54m}$Fe[20]. For these so-called "mirror nuclei", proton and neutron numbers are inverted. At and around the doubly magic, $N = Z = 28$ nucleus $^{56}$Ni, all these aspects can be observed and studied simultaneously.

The nuclear structure in the region of $^{56}$Ni is very well described by the nuclear shell model (e.g.[18]). For the decay of the 10⁺ isomer in $^{54}$Fe, experimental findings and theoretical description match remarkably well. Surprisingly, this was not the case for its mirror decay from $^{54m}$Ni[19]. In fact, in addition to its decay by electromagnetic transitions, an unexpected and significant proton-emission branch from the 10⁺ isomeric state to the first excited state of $^{53}$Co could be inferred from the γ-ray study. But despite the addition of this decay branch, shell-model theory and experiment still did not match. The discrepancy could, however, be understood by assuming that an additional proton branch of similar strength to the ground state of $^{53}$Co occurs, to which the previous experiment was insensitive to. Indeed, simple barrier-penetration calculations for each proton-emission branch show that the emission of an $E_p = 1.20$ MeV, $\ell = 5$ proton to the first excited state of $^{53}$Co is approximately as likely as the emission of an $E_p = 2.50$ MeV, $\ell = 7$ proton to its ground state[21–23]. The situation is illustrated in Fig. 1.

The present paper describes an experiment, which allowed for a four-dimensional (4D) visualization of both proton-emission branches and to derive their precise branching ratio, thus completing the picture of the decay of the 10⁺ isomer in $^{54}$Ni. The experimental results allow for the anticipated improved understanding of isospin symmetry, and provide a precision test of proton-emission theory.

## Results

The experiment was performed at the LISE3 beam line of GANIL[24]. The $^{54}$Ni ions were produced by fragmentation of a $^{58}$Ni beam at 75 MeV/nucleon on a beryllium target. The fragments were selected by the LISE3 spectrometer and implanted in the active volume of the ACTAR TPC device[25,26], a gas detector with 16384 read-out pads and an active volume of 25 cm × 25 cm × 20 cm working as a time projection chamber. Approximately 0.4% of the $^{54}$Ni ions were implanted in the 10⁺ isomeric state ($E^* = 6457$ keV; $T_{1/2} \sim 155$ ns). About half of these isomers decay by emission of a proton that can be detected in ACTAR TPC.

Due to the short half-life of the isomeric state, the ionization signal from the emitted proton is registered together with the huge signal of the ion implantation (three orders of magnitude larger), making the detection impossible with standard techniques such as silicon detectors (e.g.[11,15]). Using ACTAR TPC, the proton signal is clearly visible when the particle track projection on the collection plane creates a signal on different pads of the detector (Fig. 2-left). For each pad of the (X,Y) collection plane, the drift time of the signal, which is proportional to the Z coordinate, is measured and allows for a 3D representation of the signal distribution. The proton is emitted a short time after the implantation ($T_{1/2} \sim 155$ ns), while the ionization signal of the ion has already started to drift towards the collection plane: the time offset between the end of the ion track and the beginning of the proton track (Fig. 2-right) is then a direct measurement of the decay time of the isomeric state. The analysis of events with the

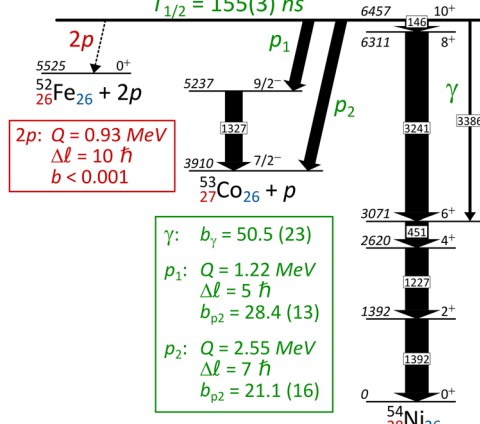

**Fig. 1 Decay scheme of $^{54m}$Ni.** The scheme is based on previous[19] and present results. Level energies are in keV and measured relative to the ground state of $^{54}$Ni. The p1 proton to the first excited state of $^{53}$Co was indirectly evidenced previously; the proton emission to the ground state (p2) is observed in the present work. The two-proton emission branch (pp) is also energetically possible, but is very unlikely because of its small energy.

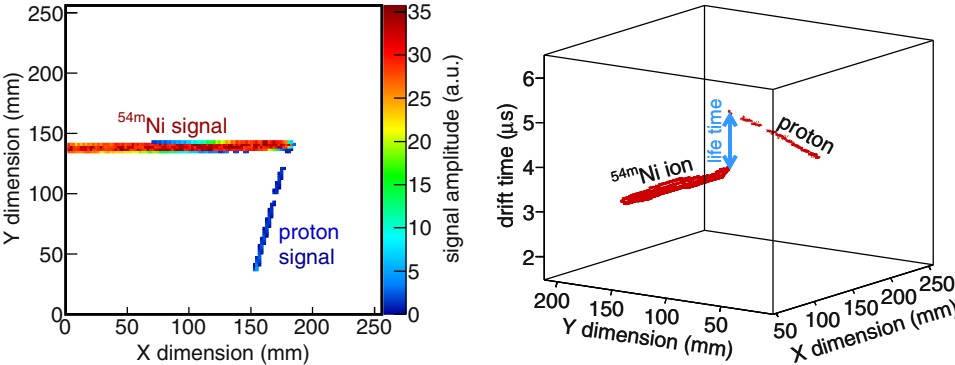

**Fig. 2 Example of a $^{54m}$Ni implantation followed by its proton radioactivity.** The left plot shows the amplitude of the ionization signal measured on the $(X, Y)$ pads of the collection plane of ACTAR TPC: the proton track can be distinguished from the ion track. In the right plot, the signal is presented in the 3D space, where the vertical coordinate is the drift time of the signal: it can be converted to the Z dimension by means of the drift velocity of the signal. The vertical gap between the two tracks is due to the decay time of the isomer: the proton is emitted while the ionization electrons created by the ion have already started to drift.

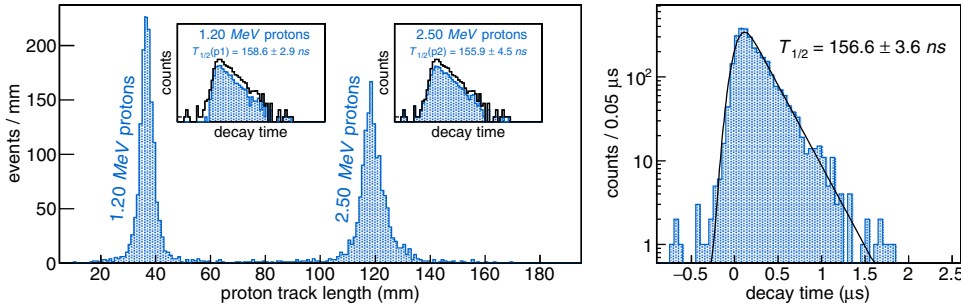

**Fig. 3 Experimental proton track length and decay time distributions.** The left plot shows the distribution of the track lengths of the protons. The right plot is the decay-time distribution obtained from the time difference measured between the ion stopping point and the proton track starting point. It is least-square fitted (black) with an exponential convoluted with a Gaussian function to account for the time-measurement resolution. The inlays of the left plot show the decay-time distributions of each proton group (blue online) compared to the total distribution (black).

implanted ion and the emitted proton thus provides 4D information about the decay: the 3D track with the related energy and the decay time of the state (Supplementary Movie 1).

During 17 h of data taking, about two million $^{54}$Ni ions were implanted in the detector. About 3000 events with proton emission were identified. The analysis of these events (ion and proton track fit) was used to build the experimental distributions of the proton track length and decay time (Fig. 3 and Supplementary Movie 2). Due to challenges associated with the charge collection plane used during the experiment, some pads regions were blind. Hence, although the detector also measures the charge collected, which is proportional to the particle energy loss (5th dimension), the determination of the proton energy from fits of the proton track length had a better resolution.

The full proton-radioactivity scheme of the isomeric state could be extracted from the analysis, with the first observation of the $\ell = 7$ proton emission at $E_p = 2.5002(43)$ MeV to the ground state of $^{53}$Co and the direct observation of the $\ell = 5$ transition at $E_p = 1.1979(44)$ MeV to the first excited state, with respectively $1411 \pm 40$ and $1459 \pm 40$ counts. The decay energies and their uncertainties were derived from refs. [19,23]. The detection efficiency was estimated from a dedicated Monte-Carlo simulation to be $(58.8 \pm 3.8)$ % for the $E_p = 1.20$ MeV proton (p1) and $(81.7 \pm 2.2)$ % for the $E_p = 2.50$ MeV proton (p2). As a result, a branching ratio of $(57.3 \pm 1.9)$ % is deduced for the $E_p = 1.20$ MeV proton with respect to the total proton emission. Combining the proton decay measured in this experiment with the previous results from γ spectroscopy[17] where the relative intensities of γ-ray emission and the $E_p = 1.20$ MeV proton branch

were determined, the absolute branching ratios of the decay of the isomeric state are $b_\gamma = (50.5 \pm 2.3)$ % and $b_p = (49.5 \pm 2.3)$ %.

As indicated in Fig. 1, a two-proton (2p) emission branch from the $^{54m}$Ni isomer into the $^{52}$Fe ground state is energetically possible. We do not have any indication of its observation in our data. This is in-line with expectations, because the barrier-penetration half-life for two-proton emission is about a factor of $10^6$ longer than for one-proton emission.

## Discussion

The combined branching ratio of both proton-emission branches, $b_p = (49.5 \pm 2.3)$ %, confirms the isospin symmetry aspects discussed in ref. [19]. The present precise result allows for an in-depth study of electromagnetic decays from the "mirror" isomers in $^{54}$Ni and $^{54}$Fe.

For the theoretical description of the high-$\ell$ proton-emission probabilities, we follow common procedures and assumptions used to calculate proton decay widths. The decay width can be factorized into a many-body nuclear structure part that gives the spectroscopic factors, $C^2S$, and a potential-barrier penetration part that gives the single-particle decay widths, $\Gamma_{sp}$:

$$\Gamma = (C^2S)\Gamma_{sp} \qquad (1)$$

The results are expressed in terms of the half-life $T_{1/2} = \hbar \ln2/\Gamma$.

We start with the $fp$ model space and then allow one proton to be excited into one of the high-$\ell$ orbitals $0h_{11/2}$ ($\ell = 5$), $0j_{13/2}$, and $0j_{15/2}$ ($\ell = 7$). The dominant proton configuration for the $10^+$ state in $^{54}$Ni is $(0f_{7/2})^7(0f_{5/2}, 1p_{3/2}, 1p_{1/2})^1$. The proton partitions are truncated to allow for the $(0f_{7/2})^7(0f_{5/2}, 1p_{3/2}, 1p_{1/2})^1$, $(0f_{7/}$

$_2)^6(0f_{5/2}, 1p_{3/2}, 1p_{1/2})^2$ or $(0f_{7/2})^6(0f_{5/2}, 1p_{3/2}, 1p_{1/2})^1(\ell)$ configurations, where $\ell$ is one of the three high-$\ell$ orbitals. The GFPX1A[27–29] and KB3G[30] Hamiltonians were used for the $fp$ shell. We use the M3Y potential[31,32] for the 182 two-body matrix elements that connect the $fp$ orbitals to the high-$\ell$ orbitals. The two-body matrix elements were calculated with harmonic-oscillator radial wave functions with $\hbar\omega = 11$ MeV. With our Woods–Saxon potential, the single-particle energy (SPE) for the proton $0h_{11/2}$ orbital is below the Coulomb plus centrifugal barrier and comes 18.7 MeV above the SPE of the $0f_{7/2}$ orbital. The wave functions of the proton $\ell = 7$ (j) orbitals are in the continuum. Based on extrapolated energies obtained by increasing the central well depth, we estimate the effective SPE of the $0j_{15/2}$ ($0j_{13/2}$) orbitals to be 32 (50) MeV above the SPE of the $0f_{7/2}$ orbital. The NuShellX code[33] is used to obtain the wave functions for $^{54}$Ni and $^{53}$Co and the proton-emission spectroscopic factors, $C^2S$.

The single-particle decay widths $\Gamma_{sp}$ were obtained from proton scattering from a Woods–Saxon potential. We start with the standard Woods–Saxon parameters used by Bohr and Mottelson[34]. The Coulomb potential was obtained from a uniform charge density distribution with radius $r_c A^{1/3}$, with $A = 53$. The parameter $r_c = 1.22$ fm was chosen to reproduce the experimental displacement energy between $^{53}$Fe and $^{53}$Co of 9.07 MeV. With the Bohr–Mottelson potential diffuseness parameter of 0.67 fm, the potential radius $r_0 = 1.26$ fm was chosen to reproduce the experimental rms charge radius of $^{52}$Fe of 3.73 fm[35]. The magnitude of the $0f_{7/2}$ proton SPE of 7.15 MeV is close to the experimental proton separation energy of $^{52}$Fe, $S_p = 7.38$ MeV. This potential was then used to calculate proton scattering from $^{53}$Co with the code WSPOT. The potential depth was adjusted to give the $Q_p$ value for each of the high-$\ell$ orbitals.

The results of calculated proton-emission probabilities are summarized in Table 1. The results with the wave functions for the GXPF1A and KB3G Hamiltonians are similar. The calculated partial half-life for the decay to the $7/2^-$ state of (0.34/0.52) μs (GFPX1A/KB3G) is in reasonably good agreement with the experimental value of $0.73 \pm 0.06$ μs. The experimental partial half-life for the decay to the $9/2^-$ state of $0.55 \pm 0.03$ μs is much smaller than those calculated. With the calculated single-particle proton decay half-life, the spectroscopic factor deduced from the experiment would be $C^2S = 4.6 \times 10^{-6}$.

The spectroscopic factor for the decay to the $9/2^-$ state from another $10^+$ state, 2 MeV higher in energy, has a spectroscopic factor of the order of $100 \times 10^{-6}$. For the decay to the $7/2^-$ state, the spectroscopic factors for these two $10^+$ states are similar. Thus, a small mixing between these two $10^+$ states would bring the theory into good agreement with experiment for the decays to both the $7/2^-$ and $9/2^-$ states. This mixing may come from a calculation in a less truncated $fp$ shell-model space, or it may reflect uncertainties in the $fp$-shell Hamiltonians.

Uncertainties for our proton-radioactivity calculations arise from (i) the fact that the potentials for the high-$\ell$ orbitals are approximated by increasing the potential depth of the Woods–Saxon potential. This changes the single-particle energies and the wave functions, which significantly modifies the spectroscopic factors and thus the branching ratios. (ii) the truncations made in the $fp$ part of the wave function, and (iii) the M3Y interaction used to connect the high-$\ell$ orbitals to the $fp$ shell. All of these approximations should be considered for improved future calculations.

It is interesting that these high $\ell$ single-particle orbitals ($\ell = 5$, 7 here, $\ell = 7$, 9 in the decay of $^{53m}$Co), which are required to mediate proton radioactivity in light nuclei near doubly magic $^{56}$Ni, are also active in super-heavy nuclei and responsible for magic numbers in these nuclei.

We have used a new method based on the 4D visualization of proton-emission events by means of a time projection chamber to study the decay of the high-lying $10^+$ isomer in $^{54}$Ni. This investigation allowed us to observe directly the decay of this isomer by proton emission to the ground and first excited states of its daughter nucleus $^{53}$Co. This observation completes the picture of the decay of this isomer, which now compares very well with its mirror decay in $^{54}$Fe.

## Methods

The ions produced by projectile fragmentation and selected by the LISE3 spectrometer were identified event by event by their energy loss ($\Delta E$) in a silicon diode located at the end of the LISE3 beamline and a time-of-flight (ToF) parameter. The latter was generated with the cyclotron radiofrequency and the timing signal of a multi-wire proportional gas counter (CFA) located just before the entrance window of ACTAR TPC signalling an ion entering the chamber.

The standard detection from the LISE3 beamline was coupled to the GET electronics[36] for ACTAR TPC, with a common dead-time as well as an event number counter and a common time stamp. The data acquisition was triggered by a signal from the CFA detector.

The ions selected by the LISE3 beamline were implanted in the ACTAR TPC device filled with a gas mixture of argon (95%) and CF₄ (5%), thus avoiding protons in the detector gas, which could generate recoil protons that could be mixed up with decay protons. The pressure was 900 mbar.

**Signal processing**. When a trigger was issued, the ionization signal from charged particles was collected on the ACTAR TPC pad plane for about 10 μs. The signal on a single pad—if above a discriminator threshold—was recorded by sampling the corresponding electronics channel with a preamplifier and shaper at 25 MHz. The processing of the raw data from the GET electronics is described in refs. [37,38], including the procedure to reconstruct the effective time distribution of the collected charge from the recorded data of each single pad. The gain and the time alignment of all electronics channels was performed using a pulse-generator signal.

The ionization signal, when amplified by the voltage on the micro-mesh of the pad plane[25], creates an induced signal on all pads, thus a distortion of the measured signal. This can be accounted for with "control channels" that are not connected to direct signal channels[38]. The pads along the beam axis were polarized in order to locally reduce the amplification[26] and prevent saturation of the electronics. This implies that the proton signals were too small to be measured on these pads.

**Event selection**. From all registered events, we were only interested in those corresponding to the implantation of $^{54m}$Ni followed by proton emission. The first selection is thus based on a simple contour in the $\Delta E$-ToF fragment identification matrix. The second selection requires the observation of the proton signal.

To identify the emitted proton, we suppressed the signal from the ion track with a dedicated algorithm: the corresponding pads were identified by an iterative search

---

**Table 1 Calculated properties of the proton-emission branches from the $10^+$ isomer in $^{54}$Ni. They are calculated with the GXPF1A and KB3G Hamiltonians.**

| $J_f$ | $n\,\ell\,j$ | $Q_p$ (MeV) | $T_{1/2,sp}$ (ps) | $C^2S$ GFPX1A | $T_{1/2}$ (μs) GFPX1A | $C^2S$ KB3G | $T_{1/2}$ (μs) KB3G |
|---|---|---|---|---|---|---|---|
| $9/2^-$ | $0h_{11/2}$ | 1.22 | 2.5 | $0.008 \times 10^{-6}$ | 200 | $0.096 \times 10^{-6}$ | 26 |
| $7/2^-$ | $0j_{13/2}$ | 2.55 | 1.3 | $2.6 \times 10^{-6}$ | 0.50 | $1.4 \times 10^{-6}$ | 0.93 |
| $7/2^-$ | $0j_{15/2}$ | 2.55 | 0.57 | $0.55 \times 10^{-6}$ | 1.04 | $0.49 \times 10^{-6}$ | 1.16 |

The columns give the spins of the orbitals contributing, their quantum numbers, the proton-emission Q values for the emission to the ground and first excited states, the partial potential-barrier penetration half-lives, the spectroscopic factors, and the partial proton-emission half-lives for the respective orbitals

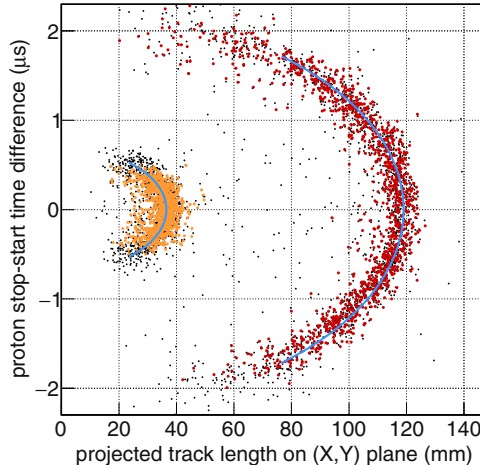

**Fig. 4 Drift velocity estimate from tracks fits.** The plot shows the distribution of the drift time difference between proton stop and start points versus the projected length of the signal on the $(X,Y)$ pad plane. The colour points are events selected for the drift velocity estimate from the 1.20 MeV (orange) and the 2.50 MeV (red) protons.

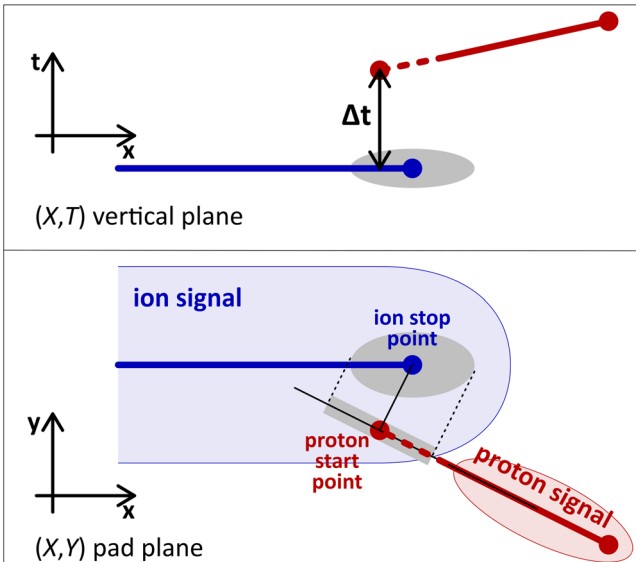

**Fig. 5 Illustration of the proton starting point estimate by track extrapolation in the fit procedure.** The starting point is defined in the $(X,Y)$ plane by the distance of the proton track extrapolation to the ion stopping point (that is non-zero due to tracks precision), and in the vertical dimension by the corresponding time. The uncertainty on the ion position (grey ellipse) converts into an uncertainty on the proton track length.

starting at the entrance side of the pad plane, with a high selection threshold. This defines the central region of the ion track, which is extended by two rows of pads all around to account for signal dispersion and pad signals below the selection threshold. Once the pads with ion signal are removed, the information from the remaining pads was analyzed to identify proton emission.

Due to missing pads in certain zones of the detector, the remaining pads were grouped into clusters, and the clusters were grouped to define particle tracks. A minimum number of pads were requested to reject noise. The event was selected if a single track can be extrapolated close to the end of the ion track (emission point). A final selection was applied based on global information about the identified track (maximum and total amplitude of the track pads, number of pads hit, track length), in order to reject remaining unwanted events such as reactions of the ion in the gas.

**Drift velocity estimate**. The event selection is finally performed by a 2D analysis of the pad signals (see Fig. 2). For a 3D analysis of the proton tracks, the drift time

information needs to be converted into the vertical $Z$ dimension by mean of the drift velocity of the ionization electrons. The identified proton signal defines a trajectory that is extrapolated to the ion stopping point, in order to estimate the projected length $L_{XY}$ on the $(X,Y)$ pad plane and the drift time difference $\Delta t$ between the proton track stop and start points. The drift velocity $v_d$ is estimated for each proton group with a linear regression defined from the total length (see Fig. 4): $L^2 = (L_{XY})^2 + (v_d \cdot \Delta t)^2$, resulting in an average value of $v_d = (53.10.4)$ mm μs$^{-1}$, in fair agreement with a GARFIELD[39] calculation.

**Track fit**. The beam direction is along the $X$ axis. The signal from pads corresponding to the ion implantation is fitted to get the ion direction angles in the $(X, Y)$ horizontal and the $(X,T)$ vertical planes. The stopping position is then defined as the last pad along $X$ with a signal, corrected with an average offset computed from the intersection of proton tracks and the beam axis.

The final analysis of the proton tracks was performed with a full 3D fit of the proton signal distribution:[38] a Bragg peak model (built from a Geant4 simulation for protons, as explained in[40] §4.4.3 with α particles) is used to compute the ionization signal along the trajectory and the 3D signal distribution is then calculated by the 3D signal dispersion along the drift towards the collection plane. Since the beginning of the proton track is hidden by the ion signal, the starting point of the proton track is the closest point of the track extrapolation in the $(X,Y)$ plane to the ion stopping position (Fig. 5).

The proton track length is then computed from the 3D fit parameters defining the trajectory. The decay time is the time difference between the ion and proton tracks at the proton start point (Fig. 2).

**Detection efficiency**. A proton emitted after the implantation of a $^{54m}$Ni ion may not be detected because its signal is hidden by the ion-track signal in the attenuation zone of the pad plane. In addition, energetic protons may escape from the active volume of ACTAR TPC. To obtain the branching ratio, $b_p$, of each proton line, it is necessary to determine the detection efficiency for each proton energy. This is achieved with a dedicated simulation built from: (1) an event generator for ion implantation and random proton-emission direction with chosen energy, (2) the simulation of the proton energy loss along its trajectory in ACTAR TPC using Geant4[41] with the gas pressure adjusted to reproduce the measured tracks length, (3) the drift of the ionization signal, with dispersion and amplification on the collection plane, (4) the processing of the signal collected on the pads.

The implantation profile of the stopping point and the ion direction distribution are defined from the ion tracks determined experimentally for all $^{54}$Ni events. The proton emission is then generated with an isotropic distribution for tracking and signal processing. Applying the same selection criteria as for the experimental data allows the determination, for each proton energy, of the global detection efficiency that combines the selection of the events as a function of the observed proton signal and the detection volume escape probability. As a large number of events can be generated, the efficiency uncertainties are dominated by systematic effects due to uncertainties of the simulation parameters (Supplementary Fig. 1).

## Data availability
The data that support the findings of this study are available from the corresponding author on reasonable request (https://doi.org/10.26143/GANIL-2019-E690).

## Code availability
The WSPOT code is available at https://people.nscl.msu.edu/~brown/reaction-codes/. The analysis codes used for the experimental data analysis are available from the corresponding author on reasonable request.

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

## Acknowledgements

We are grateful to the ion-source and accelerator staff at the Grand Accélérateur National d'Ions Lourds (GANIL) for the provision of a stable, high-intensity, primary 58Ni beam. This work was supported by the following Research Councils and Grants: European Union's Horizon 2020 Framework research and innovation programme 654002 (ENSAR2); Swedish Research Council (Vetenskapsrådet, VR 2016-3969); NSF grant PHY-1811855. B.M. is an International Research Fellow of the Japanese Society for the Promotion of Science. G.F.G. acknowledges the support of the Natural Sciences and Engineering Research Council of Canada (NSERC).

## Author contributions

B.B., D.R. and J.-C.T. prepared the proposal for the experiment, J.G., T.R., J. Pa. set up the instrumentation, B.B, J.-C.T, L.C., O.K., O.S., C.S., J.Piot, prepared the radioactive 54Ni beam, J.G., T.R., B.B., D.R., H.A.-P., A.A.R., P.A., M.C., L.C., D.M.C., B.F., J.L.F., M.G., S.G., G.F.G., B.M., A.M., J. Pa., J.Pib, J.Piot, M.V monitored the detector, data acquisition, and radioactive beam systems. J.G., T.R., B.B., D.R., and B.A.B. carried out the data analysis and interpretation of the data and prepared the manuscript.

## Competing interests

The authors declare no competing interests.
