## [Peer Review File · Nature Communications]

Reviewers' Comments:

Reviewer #1:

Remarks to the Author:

The paper "4D-imaging of drip-line radioactivity – proton emission from $54m\text{Ni}$ pictured with ACTAR TPC" by J. Giovinazzo et al. reports direct observation of two proton emission branches from the $10+$ isomeric state in the exotic proton-rich nucleus 54Ni . This measurement was achieved by imaging proton emission in the newly built time projection chamber ACTAR (TPC). The presence of one of the proton branches was deduced indirectly previously from studies of gamma rays emitted following the decay of the isomer. The observation of the second branch in this paper changed the partial gamma decay half life of the isomer bringing it in line with the half life of the corresponding isomer in the mirror nucleus 54Fe but this aspect is not discussed in the paper. The observed partial proton decay half lives were compared with theoretical calculations within the Shell Model framework and an order of magnitude agreement was found.

The proton decay probability is a product of the penetration through the Coulomb and the centrifugal barriers and the overlap between the initial and final wave functions, the so called spectroscopic factor. The first factor is relatively simple to calculate for a spherical nucleus. The spectroscopic factors were calculated using the Shell Model. These calculations are difficult since orbitals with high orbital momentum which are located high above the Fermi surface were involved in the proton emission. The calculated proton partial decay widths reproduced the data to within an order of magnitude despite the fact that the amplitudes for the orbitals under consideration were very small of the order of $\sim 10^{-6}$.

The observation of the two proton branches is unambiguous. The 54Ni nuclei produced in a fragmentation reaction were identified on event by event basis using a well established technique. Their trajectories and the trajectories of subsequently emitted protons were imaged in 3 dimensions and in time using the TPC. These images were used to extract proton energies and decay times. The paper is scientifically sound and it is well written. It reports original data. obtained with a state-of-the-art detection system. The experimental apparatus, the data analysis and the theoretical calculations are described in sufficient detail.

The structure of nuclei with exotic combination of protons and neutrons located far from the line of stability is one of the focal points of the contemporary nuclear physics. The proton decay combines aspects of quantum tunneling with structure of very exotic nuclei. The 54Ni nucleus is only a second case of proton emission from a nucleus with $Z < 50$ albeit from an isomeric state. Consequently, the paper is of interest for nuclear physicists and possibly for a more general audience.

I would encourage the authors to take into account the following suggestions in order to improve the paper.

1. There is a vast literature on the proton decay in heavier nuclei with $Z > 50$ and on various models of the proton decay. It would be beneficial for the reader if at least some reference(s) to review articles on the subject were included in the paper.
2. The proton decay width is very sensitive to the proton energy. It is not clear to me what is the value and accuracy of the proton energy determined in the paper. Was this value used in the calculations? If the literature value was used instead it should be quoted in the paper together with the error. The authors should also comment on the impact of the energy accuracy on the calculations.
3. The authors state in the abstract that the data are best explained by the shell model. That implies that several models were compared in the paper which is not the case. To avoid confusion, I would replace "best explained" by "explained" in the abstract.
4. The levels appear to be missing in Figure 1.

Reviewer #2:

Remarks to the Author:

Key results

The study of proton radioactivity can be a powerful tool to shed light into different aspects of nuclear structure away from the beta-stability valley. The work by Giovanazzo et al., "4D-imaging of drip-line radioactivity - proton emission from ^{54m}Ni pictured with ACTAR TPC" addresses this topic by exploiting the possibility to look at the time-evolution of 3D-reconstructed nuclear decays trajectories given by the time-projection chamber. In particular, the authors re-investigated the decay properties of the proton-unbound $10+$ isomer in ^{54}Ni , in order to address the question of the differences with respect to its mirror partner, ^{54m}Fe . In fact, if previously the nuclear structure in this region around ^{56}Ni had seemed to be understood, including ^{54m}Fe , this was not the case for ^{54m}Ni decay. In addition to de-exciting by gamma emission, as does ^{54m}Fe , ^{54m}Ni presented also an important proton-decay branch and a disagreement between experiment and theory. In their work, the authors observed ^{54m}Ni to decay by emission of two proton transitions, which carry unusually-high angular momentum, feeding the ground- and the first-excited state in the daughter nucleus, respectively. The former decay branch was observed in this work for the first time and the branching ratios accurately determined, allowing to explain the structure of this nuclear isomer. The results presented by the authors in their work help probing proton-emission theories and better understanding isospin symmetry.

Validity and significance

The work discusses data that are original. As far as I can tell, the methodology is sound and the data treatment is valid. The team is well known for its experience and expertise in such types of experiments. The results are of interest for the nuclear physics community.

Data and methodology

The experimental methodology used in the work presented by the authors is in part standard and in part new, in both cases it is adequate. The isotope production and separation technique is well known and standard for such studies. On the other hand, even though time-projection chambers have been under development and used in decay-studies for a few years now, to my knowledge it is the first time that such an analysis could and was conducted, with the time-axis added to the 3D-reconstruction of events. The data appear to be of very good quality and statistical significance, with apparently no background and low systematic uncertainty. The data selection and analysis methodology is well described and properly shared between the main paper and the methods section. Nevertheless, I do have a few questions, which are detailed below.

Clarity and context

The manuscript is well written, concise and clear also for non-experts. It is in my opinion suitable for publication after the comments below are taken into account.

References

The bibliography is appropriate.

Suggested improvements

I would like to ask the authors to address the following questions before recommending the manuscript for publication.

Figure 1: The figure would be more easily readable with a more traditional graphics having the levels drawn at the bottom of the arrows.

Introduction: when the ^{54m}Fe isomer is introduced for the first time a reference should be given and

its spin and parity as well.

Results and method sections:

- it would be helpful to know which other nuclear species were present in the cocktail beam entering the chamber and if any of them could cause background events.

- It should be explicitly stated whether the authors observed any ion-drifting, i.e. whether the observed distribution of the protons direction was compatible with an isotropic distribution, as assumed in the simulations.

- In the last sentence of the drift-velocity estimate section, it isn't clear to me what "in agreement with expectations" refers to. Do the "expectations" refer to calculated/simulated drift velocity or to measurements carried on separately for this gas mixture and meteo conditions? In the reference [Gio20], the authors show that drift-velocity simulations performed with the GARFIELD code are not performing too well with a P10-based mixture and no explanation could be found for it. Is this the case also for the Ar/CF4 mixture used in the work described in the manuscript? In any case, it is good that the data analysis could benefit from a direct measurement of the drift velocity under the experiment conditions and did not need to rely on simulations.

- In the discussion of the track fitting procedure, a Bragg peak model is mentioned. A reference should be given to my understanding to GEANT (see ref. [2] in ref. [Gio20]). A comment on whether the track lengths are indeed in agreement with the range of 1.2 and 2.5 MeV protons in the given gas mixture should also be added.

- In the detection efficiency section, in the last sentence systematic effects due to the uncertainties of the simulation parameters are given as the main source of uncertainty. What is the order of such uncertainties?

Discussion: in table 1 results from only one of the two hamiltonians considered are presented and a statement is made in the text that the results are similar. I would recommend to add the results from calculations with the KB3G hamiltonian to the table or at least quote within which margin the results of the calculations are similar.

A short global conclusion/summary at the end of the paper is missing.

Reviewer #3:

Remarks to the Author:

In the present manuscript, the Authors report the experimental measurements of proton-emission branches from an isomeric state in ^{54}mNi , visualized in a newly developed detector. They claim that these decays, which carry an unusually high angular momentum, $\ell = 5$ and $\ell = 7$, respectively, are best explained with a potential model for the proton barrier penetration and a shell-model calculation for the overlap of the initial and final wave functions.

The experimental results presented are very interesting and are quite well explained in the text. I just would like to ask the Authors to make a figure like the right plot of fig 3, with the decay time distribution of proton p1, and a similar one for the proton p2, in order to see the compatibility of the two half-lives, and be sure that the decay is from the same state.

However, the theoretical interpretation of the experimental results presented in the manuscript needs a revision.

The Authors use a theoretical approach where the single particle decay states are obtained from a

potential model and a shell-model calculation gives the spectroscopic factors as the overlap of the initial and final wave functions. The half lives are than calculated in a standard way.

The single particle energies that were used in the shell model calculation for the $h_{11/2}$, $j_{15/2}$, and $j_{13/2}$ states, as 22MeV, 44MeV, and 54 MeV respectively, seem to be very high in energy. I would like to ask the Authors to calculate with program WSPOT, referred in the text, the energy and width of the resonances of the $h_{11/2}$, $j_{13/2}$ and $j_{15/2}$ states. This could give an indication for the energies of these states.

The results of their calculation of the spectroscopic factors and half lives are presented in Table 1. It is interesting that the combined proton emission half life is reproduced well, however, the same cannot be said of the partial results.

In these studies, if one looks separately at half- lives, it might be difficult to arrive at solid conclusions, since the results depend on many parameters, as also the Authors acknowledge. More reliable conclusions can be derived by looking at ratios, where this dependence mainly .drops off. The branching ratios reported in Table 1, are 8% and 92% for the first excited and ground states respectively. The ratio between these quantities is .087. These values are quite off the values obtained from experiment, where the ratio between 57% and 43% gives 1.32, corresponding to a factor of 15 of difference between theory and experiment in quantities that are not very dependent on parameters.

This is not at all a "similar strength" that "should be viewed as a big success" as reported in the manuscript.

In fact, it is a quite far from being successful.

Some of the sentences in the text should also be improved.

For example, in line 37 of the Abstract, it cannot be said that the decays "are best explained" There is no comparison with any other model, and the agreement between theory and experiment is poor.

In line 61 it is not only the Coulomb barrier, but also the centrifugal one.

It would be useful for the reader to explain in line 90 why the previous experiment was insensitive to the proton branch.

The use here of "two proton emission" in line 91 can be confused with the simultaneous emission of two protons.

In line 212, it should be added that there is also a dependence on the single particle energies and wave functions of the high l orbitals.

About the comment on line 227, I do not find it so amazing when compared to the fact that 50 years ago, proton emission was discovered as emission from an even higher single particle orbit $l=9$ in ^{53}Co . That is probably the only nucleus, where a decay from an $l=9$ state will be observed, so, it is really quite amazing that the $l=9$ might be only important for ^{53}Co .

From the above discussion I consider that the manuscript should not be published in its present state.

In our answer, we used the following convention

- General comments from the reviewers / editors
- **Question from the reviewers or points to be addressed**
- Answers from the authors

Reviewer #1 (Remarks to the Author):

The paper “4D-imaging of drip-line radioactivity – proton emission from 54mNi pictured with ACTAR TPC” by J. Giovinazzo et al. reports direct observation of two proton emission branches from the 10+ isomeric state in the exotic proton-rich nucleus 54Ni . This measurement was achieved by imaging proton emission in the newly built time projection chamber ACTAR (TPC). The presence of one of the proton branches was deduced indirectly previously from studies of gamma rays emitted following the decay of the isomer. The observation of the second branch in this paper changed the partial gamma decay half life of the isomer bringing it in line with the half-life of the corresponding isomer in the mirror nucleus 54Fe but this aspect is not discussed in the paper. The observed partial proton decay half-lives were compared with theoretical calculations within the Shell Model framework and an order of magnitude agreement was found.

The proton decay probability is a product of the penetration through the Coulomb and the centrifugal barriers and the overlap between the initial and final wave functions, the so-called spectroscopic factor. The first factor is relatively simple to calculate for a spherical nucleus. The spectroscopic factors were calculated using the Shell Model. These calculations are difficult since orbitals with high orbital momentum which are located high above the Fermi surface were involved in the proton emission. The calculated proton partial decay widths reproduced the data to within an order of magnitude despite the fact that the amplitudes for the orbitals under consideration were very small of the order of $\sim 10^{-6}$.

The observation of the two proton branches is unambiguous. The 54Ni nuclei produced in a fragmentation reaction were identified on event by event basis using a well established technique. Their trajectories and the trajectories of subsequently emitted protons were imaged in 3 dimensions and in time using the TPC. These images were used to extract proton energies and decay times. The paper is scientifically sound and it is well written. It reports original data obtained with a state-of-the-art detection system. The experimental apparatus, the data analysis and the theoretical calculations are described in sufficient detail.

The structure of nuclei with exotic combination of protons and neutrons located far from the line of stability is one of the focal points of the contemporary nuclear physics. The proton decay combines aspects of quantum tunneling with structure of very exotic nuclei. The 54Ni nucleus is only a second case of proton emission from a nucleus with $Z < 50$ albeit from an isomeric state. Consequently, the paper is of interest for nuclear physicists and possibly for a more general audience.

I would encourage the authors to take into account the following suggestions in order to improve the paper.

1. There is a vast literature on the proton decay in heavier nuclei with $Z > 50$ and on various models of the proton decay. It would be beneficial for the reader if at least some reference(s) to review articles on the subject were included in the paper.

We have added the references to two of the latest review articles on proton radioactivity (in the introduction section).

2. The proton decay width is very sensitive to the proton energy. It is not clear to me what is the value and accuracy of the proton energy determined in the paper. Was this value used in the calculations? If the literature value was used instead it should be quoted in the paper together with the error. The authors should also comment on the impact of the energy accuracy on the calculations.

The proton energies rely on high precision mass measurements of both ^{54}Ni (mother) [1] and ^{53}Co (daughter) [2] as well as the previous gamma-ray study of ^{54}Ni [3]. These experiments are much more precise as far as decay energies and thus E_p and Q_p values are concerned.

The present best values of $E_p = 2.5002(43)$ MeV for the transition to the ground state of ^{53}Co and for the transition into the excited state with $E_p = 1.1979(44)$ MeV are now presented with full value and uncertainty in the section RESULTS of the revised manuscript. In the INTRODUCTION we added the Zho17 reference but prefer to keep the values 2.50 MeV and 1.20 MeV there to ease readability.

One may add (see also response to reviewer 3) that the few-keV uncertainties in decay energy play a minor role if any in the theoretical assessment.

[1] P. Zhong et al., Phys. Lett. B 767, 20 (2017). Added to revised manuscript as [Zho17].

[2] A. Kankainen et al., Phys. Rev. C 82, 034311 (2010). [Kan10] in original manuscript.

[3] D. Rudolph et al., Phys. Rev. C 78, 021301(R) (2008). [Rud08] in original manuscript.

3. The authors state in the abstract that the data are best explained by the shell model. That implies that several models were compared in the paper which is not the case. To avoid confusion, I would replace “best explained” by “explained” in the abstract.

We modified the text according to reviewer suggestion.

4. The levels appear to be missing in Figure 1.

The levels seem to appear or not depending of the computer/software. The figure has been recreated with another drawing software.

In our answer, we used the following convention

- General comments from the reviewers / editors
- **Question from the reviewers or points to be addressed**
- Answers from the authors

Reviewer #2 (Remarks to the Author):

Key results

The study of proton radioactivity can be a powerful tool to shed light into different aspects of nuclear structure away from the beta-stability valley. The work by Giovinazzo et al., "4D-imaging of drip-line radioactivity - proton emission from ^{54m}Ni pictured with ACTAR TPC" addresses this topic by exploiting the possibility to look at the time-evolution of 3D-reconstructed nuclear decays trajectories given by the time-projection chamber. In particular, the authors re-investigated the decay properties of the proton-unbound $10+$ isomer in ^{54}Ni , in order to address the question of the differences with respect to its mirror partner, ^{54m}Fe . In fact, if previously the nuclear structure in this region around ^{56}Ni had seemed to be understood, including ^{54m}Fe , this was not the case for ^{54m}Ni decay. In addition to de-exciting by gamma emission, as does ^{54m}Fe , ^{54m}Ni presented also an important proton-decay branch and a disagreement between experiment and theory. In their work, the authors observed ^{54m}Ni to decay by emission of two proton transitions, which carry unusually-high angular momentum, feeding the ground- and the first-excited state in the daughter nucleus, respectively. The former decay branch was observed in this work for the first time and the branching ratios accurately determined, allowing to explain the structure of this nuclear isomer. The results presented by the authors in their work help probing proton-emission theories and better understanding isospin symmetry.

Validity and significance

The work discusses data that are original. As far as I can tell, the methodology is sound and the data treatment is valid. The team is well known for its experience and expertise in such types of experiments. The results are of interest for the nuclear physics community.

Data and methodology

The experimental methodology used in the work presented by the authors is in part standard and in part new, in both cases it is adequate. The isotope production and separation technique is well known and standard for such studies. On the other hand, even though time-projection chambers have been under development and used in decay-studies for a few years now, to my knowledge it is the first time that such an analysis could and was conducted, with the time-axis added to the 3D-reconstruction of events. The data appear to be of very good quality and statistical significance, with apparently no background and low systematic uncertainty. The data selection and analysis methodology is well described and properly shared between the main paper and the methods section. Nevertheless, I do have a few questions, which are detailed below.

Clarity and context

The manuscript is well written, concise and clear also for non-experts. It is in my opinion suitable for publication after the comments below are taken into account.

References

The bibliography is appropriate.

Suggested improvements

I would like to ask the authors to address the following questions before recommending the manuscript for publication.

Figure 1: The figure would be more easily readable with a more traditional graphics having the levels drawn at the bottom of the arrows.

There is indeed a problem with the figure file. The figure has been recreated with another drawing software.

Introduction: when the ^{54m}Fe isomer is introduced for the first time a reference should be given and its spin and parity as well.

The sentence is modified to include “ 10^+ ” and reference [Daf78] is added: It reads ... well-studied 10^+ isomer ^{54m}Fe [Daf78]. The change “well-known” to “well-studied” is motivated by the fact that [Daf78], which includes the measurement of electromagnetic moments, was published prior to the conference proceedings, in which the group reported on the observation of the isomer as such [Noe78]. We prefer to reference to the journal article, because the proceedings are very difficult to find; [Noe78] is cited “in press” as first reference in [Daf78].

[Daf78] Dafni, E., Noé, J.W., Rafailovich, M.H., and Sprouse, G.D., Static moments of $^{54}\text{Fe}^m$ and perturbed angular distributions with combined dipole and quadrupole interactions, *Phys. Lett.* 76B, 51-53 (1978).

[Noe78] Noé, J.W., Geesaman, D.F., Gural, P., and Sprouse, G.D., Proc. Int. Conf. on Medium-Light Nuclei, Florence, Italy, 1977, eds. Blasi, P., and Ricci, R.A., 458 (1978).

Results and method sections:

- **it would be helpful to know which other nuclear species were present in the cocktail beam entering the chamber and if any of them could cause background events.**

Indeed, the ^{54}Ni fragments are selected with a few other species that also reach the detector: mainly ^{49}Mn , ^{51}Fe , $^{53,52}\text{Co}$, and ^{56}Cu . Together, these represent about 55% of the fragments. They are β emitters with half-lives in the 90-300 ms range. As tuned in the experiment, the detector is insensitive to β particles, with only a small β -delayed proton branch for ^{56}Cu of about 0.4%.

The only background could come from the β -p decay of ^{56}Cu or proton emission from ^{53m}Co ($T_{1/2} \sim 250$ ms) – that has also been studied in this experiment. This is extremely unlikely for the following reasons, illustrated for the case of the ^{53m}Co proton decay:

- the half-life of ^{54m}Ni is very short: the proton is registered in the same event as the implantation; a random coincidence of ^{54}Ni (ground or isomeric state) with a proton from ^{53m}Co decay needs to occur in the few μs time window after implantation (the recording/sampling time is about 10 μs : the effective coincidence time is a bit less since the ion signal comes about 2 μs after the beginning of the sampling window);
- ^{53}Co production is about 8% of the cocktail beam, and considerably less than 1% is produced in the isomeric state;
- the total implantation rate was few tens ions per second (less than 100 Hz);
- from these (overestimated) numbers, the probability of one random coincidence is in the order of $5 \cdot 10^{-7}$, so a total contamination of about 1 events (with $2.5 \cdot 10^6$ implanted ions of ^{54}Ni) out of 2500 proton emissions from ^{54m}Ni ;

- in addition, it would require also that the ^{53m}Co and the ^{54}Ni fragments are implanted in the same area (within few mm) for proton (from ^{53m}Co) of the random coincidence to be considered as emitted from ^{54}Ni ;
- finally, the proton decay energies of ^{53m}Co and ^{54m}Ni are different.

Numbers are of the same order in the case of ^{56}Cu β -p decay: 2.4% production, 0.4% proton branching. From these considerations, we conclude that our analysis is free of contamination from other nuclei produced in this setting.

- **It should be explicitly stated whether the authors observed any ion-drifting, i.e. whether the observed distribution of the protons direction was compatible with an isotropic distribution, as assumed in the simulations.**

Ion drift has been clearly identified in the case of ^{53m}Co proton emission, for a fraction of the implanted fragments (from the angular distribution and the signal dispersion that varies with the drift length). Since in that case the half-life is much larger (~ 250 ns), a non-neutralized ion drifts completely to the cathode.

The drift velocity of ions is about 1000 times slower (see ref. below Sch77, Yam89, Gol15) than the one of the electrons, in the order of $5 \cdot 10^{-2}$ mm/ μs . For an average drift time of 150 ns (1 half-life), this corresponds to $7.5 \cdot 10^{-3}$ mm, inducing an error (overestimation) of ~ 0.15 ns of the half-life (considering all ions are drifting). This is far below the uncertainty from the track analysis (3.6 ns). This is consistent with figure 4 where we do not observe a reduced number of events going towards the pad-plane (negative stop-start time difference) with respect to events going towards the cathode (positive stop-start time difference).

- [Sch77] G. Schultz, G. Charpak, F. Sauli, *Mobilities of positive ions in some gas mixtures used in proportional and drift chambers*, Revue de Physique Appliquée, 1977, tome 12, p. 67
- [Yam89] T. Yamashita et al., *Measurements of the electron drift velocity and positive-ion mobility for gases containing CF₄*, Nucl. Instr. And Meth. In Phys. Res. A283 (1989) 709-715
- [Gol15] R.I. Golyatina, S.A. Maiorov, *Approximation of ion drift velocity in own gas*, Bulletin of the Lebedev Physics Institute, 2015, Vol. 42, No. 10, pp. 294-298

- **In the last sentence of the drift-velocity estimate section, it isn't clear to me what "in agreement with expectations" refers to. Do the "expectations" refer to calculated/simulated drift velocity or to measurements carried on separately for this gas mixture and meteo conditions? In the reference [Gio20], the authors show that drift-velocity simulations performed with the GARFIELD code are not performing too well with a P10-based mixture and no explanation could be found for it. Is this the case also for the Ar/CF₄ mixture used in the work described in the manuscript? In any case, it is good that the data analysis could benefit from a direct measurement of the drift velocity under the experiment conditions and did not need to rely on simulations.**

The agreement here is with respect to the GARFIELD calculation that gives a drift velocity of ~ 56 mm/ μs for a field of 100 V/cm used in the experiment, which is in that case in relatively good agreement with the value estimated empirically from the protons tracks. The fact that in this case the agreement with GARFIELD is good while it was not in ref. [Gio20] is not an issue here, because:

1. we can measure it from the data, in the experiment conditions, so *we use this value*;
2. in ref. [Gio20], it was a different gas: P10;
3. more systematic studies would be interesting (see for example [Col02]), but this would not affect the results of this paper.

We modified the text from “resulting in an average value of $v_d = (53.1 \pm 0.4) \text{ mm}/\mu\text{s}$, in fair agreement with expectations” to **“resulting in an average value of $v_d = (53.1 \pm 0.4) \text{ mm}/\mu\text{s}$, in fair agreement with a GARFIELD [ref] calculation”**, with a reference to the GARFIELD program.

[Col02] P. Colas et al., Electron drift velocity measurements at high electric fields, *Nucl. Instr. and Meth. In Phys. Res. A* 478 (2002) 215-219

- **In the discussion of the track fitting procedure, a Bragg peak model is mentioned. A reference should be given to my understanding to GEANT (see ref. [2] in ref. [Gio20]). A comment on whether the track lengths are indeed in agreement with the range of 1.2 and 2.5 MeV protons in the given gas mixture should also be added.**

We included the reference [Gio18] (§4.4.3) where the method is proposed for alpha particles instead of protons. The agreement of track length has no point here: in the case of experimental tracks fitting, the “Bragg peak model” is only a normalized (in length and amplitude) pattern, tuned by two parameters to fit to the shape of the measured signal distribution between the starting and stopping point. The length is not deduced from this pattern, but from the fitted start/stop points. **We did not add a full explanation of the fit model as we think it is too much detailed for the scope of the paper** and would make the reading less easy. A dedicated “technical” paper will be more suitable.

Nevertheless, the agreement between the track length from the measurement and from the simulation (for efficiency estimates) needs to be fulfilled. Since the energies of the protons are known with enough precision, the gas pressure used in the simulation has been adjusted to this purpose (a systematic uncertainty is considered, by varying the simulation pressure of $\pm 20 \text{ mbar}$, values for which the agreement is clearly lost – see also the answer to next comment on systematic uncertainties).

We added the following precision in the “Detection efficiency” section: “[...] the simulation of the proton energy loss along its trajectory in ACTAR TPC using Geant4 [Ago03] **with the gas pressure adjusted to reproduce the measured tracks length**”.

[Gio18] Giovanazzo, J. et al., Metal-core pad-plane development for ACTAR TPC, *Nucl. Instrum. Methods Phys. Res. A* 892, 114-121 (2018)

- **In the detection efficiency section, in the last sentence systematic effects due to the uncertainties of the simulation parameters are given as the main source of uncertainty. What is the order of such uncertainties?**

The error budget for the efficiency estimate from the simulation is the following (the main terms differ for protons p1 and p2 due to the difference in track length: about 35 and 115 mm):

uncertainty source	p1 (1.20 MeV)	p2 (2.50 MeV)
statistical	$\pm 0.29\%$	$\pm 0.30\%$
systematics		
beam suppression size	$\pm 2.52\%$	$\pm 0.54\%$
pressure	$\pm 1.19\%$	$\pm 1.54\%$
analysis threshold	$\pm 0.50\%$	$\pm 0.01\%$
pads multiplicity validation	$\pm 2.52\%$	$\pm 0.26\%$
escape pads	$\pm 0.15\%$	$\pm 1.34\%$
signal collection	$\pm 0.39\%$	$\pm 0.01\%$
electronics amplification	$\pm 0.22\%$	$\pm 0.01\%$
output noise	$\pm 0.14\%$	$\pm 0.01\%$
input noise	$\pm 0.01\%$	$\pm 0.01\%$

The variations of the efficiencies for both proton energies are presented in the figure below; the dashed lines represent the statistical uncertainty while the dotted line is the total uncertainty (statistical and systematics). The correlations between parameters variation have been studied, but the impact on the total error being small, they were not considered in the final uncertainties. The gas pressure in the simulation is adjusted to reproduce the observed tracks length. The figure also displays results with other usual electromagnetic physics lists (the pressure is adjusted for each physics list on the track length). **We suggest adding the following picture as additional material to the publication.**

Summary of the systematic error terms for the detection efficiency estimate from the simulation, for 1.20 MeV protons (upper plot) and 2.50 MeV protons (lower plot) emitted by ^{54m}Ni . The full line shows the estimated efficiency, the dashed lines the statistical uncertainty (from the number of simulated events) and the dotted lines the total uncertainty including: the physics list, the beam (ions) signal suppression size, the gas pressure, the pads analysis threshold, the minimum protons signal pads multiplicity, the number of border pads to consider escaping protons, the signal collection amplification factor, the electronic channels amplification, the electronics output noise level and the signal input noise level.

Discussion: in table 1 results from only one of the two hamiltonians considered are presented and a statement is made in the text that the results are similar. I would recommend to add the results from calculations with the KB3G hamiltonian to the table or at least quote within which margin the results of the calculations are similar.

We have now added the results from both Hamiltonians to the table and both half-life results to the text. As can be seen now, there is only little difference between the two calculations as far as they can be obtained in our calculations.

A short global conclusion/summary at the end of the paper is missing.

Although not necessarily customary in *Nature Communications*, we have added three sentences of conclusions.

In our answer, we used the following convention

- General comments from the reviewers / editors
- **Question from the reviewers or points to be addressed**
- Answers from the authors

Reviewer #3 (Remarks to the Author):

In the present manuscript, the Authors report the experimental measurements of proton-emission branches from an isomeric state in ^{54}mNi , visualized in a newly developed detector. They claim that these decays, which carry an unusually high angular momentum, $\ell = 5$ and $\ell = 7$, respectively, are best explained with a potential model for the proton barrier penetration and a shell-model calculation for the overlap of the initial and final wave functions.

The experimental results presented are very interesting and are quite well explained in the text. **I just would like to ask the Authors to make a figure like the right plot of fig 3, with the decay time distribution of proton p1, and a similar one for the proton p2, in order to see the compatibility of the two half-lives, and be sure that the decay is from the same state.**

We show here the plots of the decay times for protons p1 and p2. The half-life fit results are the following:

- All: $T_{1/2} = 156.6 \pm 3.6 \text{ ns}$
- P1: $T_{1/2} = 158.6 \pm 2.9 \text{ ns}$
- P2: $T_{1/2} = 155.9 \pm 4.5 \text{ ns}$

The average half-life from P1 and P2 fits is $157.8 \pm 2.5 \text{ ns}$. The larger uncertainty from the global fit is due to the uncertainty caused by the few counts at short times that are mainly present in the P2 protons. While all these values are compatible, we keep the global fit that contains all information.

We modified the figure 3 by the following one, where the inlays show the decay time distributions of

the p1 and p2 protons groups (the caption is modified accordingly).

However, the theoretical interpretation of the experimental results presented in the manuscript needs a revision.

The Authors use a theoretical approach where the single particle decay states are obtained from a potential model and a shell-model calculation gives the spectroscopic factors as the overlap of the initial and final wave functions. The half-lives are then calculated in a standard way.

The single particle energies that were used in the shell model calculation for the $h_{11/2}$, $j_{15/2}$, and $j_{13/2}$ states, as 22MeV, 44MeV, and 54 MeV respectively, seem to be very high in energy. I would like to ask the Authors to calculate with program WSPOT, referred in the text, the energy and width of the resonances of the $h_{11/2}$, $j_{13/2}$ and $j_{15/2}$ states. This could give an indication for the energies of these states.

The referee's suggestion about using WSPOT to obtain the single-particle energies (SPE) is very good. We are able to calculate SPE for unbound states up to the top of the Coulomb plus centrifugal barrier - about 18 MeV. Thus, we studied the systematics of the SPE as a function of the well depth. The attached figure shows the results for some high-j orbitals with $N=1$ being the potential described in the text. In that case, only the $0h_{11/2}$ orbital can be obtained and it turns out to be 3.3 MeV lower than our harmonic oscillator estimate (shown by the blue cross). The results for the $l=7$ orbital (j) have to be extrapolated. To do this, we look at the SPE obtained by multiplying the central well depth of the potential by a factor N . The results shown in the figure below indicate that the SPE are slightly bent down at higher energy (smaller N) compared to a linear extrapolation from lower energy (larger N). We have drawn dashed lines to indicate an estimated extrapolation to $N=1$. The results is that the $0j_{15/2}$ SPE is 5.3 MeV lower than our harmonic oscillator estimate and the $0j_{13/2}$ SPE is 0.7 MeV higher than our harmonic-oscillator plus 10 MeV spin-orbit estimate.

However, while redoing all shell-model calculations, we found also a problem with the calculation of the spectroscopic factors for the decay to the first excited state. In fact, the results are very sensitive to even very small modifications to the input parameters. We found that the detailed structure of the 10^+ state is very important. There is a 10^+_{3} state, just 2 MeV above the 10^+_{1} state, which plays a major role for our results. The spectroscopic factors for this 10^+_{3} state are very large for both Hamiltonians and a small change in mixing between the two states greatly modifies our result for the 10^+ state of interest. We have changed to the text to discuss this situation.

The results of their calculation of the spectroscopic factors and half-lives are presented in Table 1. It is interesting that the combined proton emission half-life is reproduced well, however, the same cannot be said of the partial results.

In these studies, if one looks separately at half-lives, it might be difficult to arrive at solid conclusions, since the results depend on many parameters, as also the Authors acknowledge.

More reliable conclusions can be derived by looking at ratios, where this dependence mainly drops off. The branching ratios reported in Table 1, are 8% and 92% for the first excited and ground states respectively. The ratio between these quantities is .087

These values are quite off the values obtained from experiment, where the ratio between 57% and 43% gives 1.32, corresponding to a factor of 15 of difference between theory and experiment in quantities that are not very dependent on parameters.

This is not at all a “similar strength” that “should be viewed as a big success” as reported in the manuscript. In fact, it is a quite far from being successful.

In view of what we said to the previous comment of the referee, this discussion was removed from the paper.

Some of the sentences in the text should also be improved.

For example, in line 37 of the Abstract, it cannot be said that the decays “are best explained” There is no comparison with any other model, and the agreement between theory and experiment is poor.

We have replace “best explained” by “can be approximated theoretically”, however, we insist again on the fact that to the best of our knowledge no other model was able to get an agreement at the same level as our calculations, might it only be because too few people tried.....

In line 61 it is not only the Coulomb barrier, but also the centrifugal one.

The text is modified: “[...] by the Coulomb and centrifugal barrier”.

It would be useful for the reader to explain in line 90 why the previous experiment was insensitive to the proton branch.

In the RISING collaboration experiment [Rud08], the nuclei produced by fragmentation were implanted in a passive stopper, surrounded by gamma detectors. There was no charged particle detector at the implantation point. Even if the ions would have been implanted in an active stopper (a silicon detector), it would have been impossible to observe the ^{54m}Ni decay protons: the energy deposit of the proton is 3 order of magnitude lower than the energy deposit of the implantation, and since it occurs about 150 ns after, the proton signal would have been hidden by the ion signal at the preamplifier stage. This was the reason to perform the experiment with a TPC.

We think the explanation is too complicated in this introduction and propose not to add it. This is anyway mentioned in the second paragraph of the “Results” section.

The use here of “two proton emission” in line 91 can be confused with the simultaneous emission of two protons.

The text is modified in order to avoid this confusion.

In line 212, it should be added that there is also a dependence on the single particle energies and wave functions of the high l orbitals.

We added this to the text.

About the comment on line 227, I do not find it so amazing when compared to the fact that 50 years ago, proton emission was discovered as emission from an even higher single particle orbit l=9 in ^{53}Co . That is probably the only nucleus, where a decay from an l=9 state will be observed, so, it is really quite amazing that the l=9 might be only important for ^{53}Co .

We rephrased this sentence including also the l=9 transition in $^{53\text{m}}\text{Co}$ in the discussion. However, we also precise that the aim of this sentence was not to discuss either $^{53\text{m}}\text{Co}$ or $^{54\text{m}}\text{Ni}$ decay, but rather to underline that these high-l orbitals play a major role also for our understanding of the structure of very heavy and super-heavy nuclei.

From the above discussion I consider that the manuscript should not be published in its present state.

We hope that with the modifications introduced in the manuscript, the paper is now ready to be published.

Reviewers' Comments:

Reviewer #1:

Remarks to the Author:

The authors included my suggestions in the current version of the paper. Also, the modified discussion of the shell model calculations points to a plausible explanation of the discrepancy between the calculated and measured proton branching ratios. In my opinion, the paper is ready for publication in its current form.

Reviewer #2:

Remarks to the Author:

The authors have addressed my concerns and questions. I therefore recommend the manuscript for publication.

Reviewer #3:

Remarks to the Author:

I consider that the Authors have answered the question satisfactorily, so I recommend the manuscript for publication.